# The effect of caloric restriction on the increase in senescence-associated T cells and metabolic disorders in aged mice

Xiaoxiang Yan[1,2], Natsumi Imano[3,4], Kayoko Tamaki[2,3], Motoaki Sano[2], Ken Shinmura[2,3]*

**1** Ruijin Hospital, Institute of Cardiovascular Diseases and Department of Cardiology, Shanghai Jiao Tong University School of Medicine, Shanghai, PR China, **2** Department of Cardiology, Keio University School of Medicine, Tokyo, Japan, **3** Department of General Internal Medicine, Hyogo College of Medicine, Nishinomiya, Japan, **4** Department of Bioscience, School of Science and Technology, Kwansei Gakuin University, Sanda, Japan

* ke-shimmura@hyo-med.ac.jp, shimmura@keio.jp

**Data Availability Statement:** All relevant data are within the paper and its Supporting Information files.

## Abstract

Aging is associated with functional decline in the immune system and increases the risk of chronic diseases owing to smoldering inflammation. In the present study, we demonstrated an age-related increase in the accumulation of Programmed Death-1 (PD-1)+ memory-phenotype T cells that are considered "senescence-associated T cells" in both the visceral adipose tissue and spleen. As caloric restriction is an established intervention scientifically proven to exert anti-aging effects and greatly affects physiological and pathophysiological alterations with advanced age, we evaluated the effect of caloric restriction on the increase in this T-cell subpopulation and glucose tolerance in aged mice. Long-term caloric restriction significantly decreased the number of PD-1+ memory-phenotype cluster of differentiation (CD) 4+ and CD8+ T cells in the spleen and visceral adipose tissue, decreased M1-type macrophage accumulation in visceral adipose tissue, and improved insulin resistance in aged mice. Furthermore, the immunological depletion of PD-1+ T cells reduced adipose inflammation and improved insulin resistance in aged mice. Taken together with our previous report, these results indicate that senescence-related T-cell subpopulations are involved in the development of chronic inflammation and insulin resistance in the context of chronological aging and obesity. Thus, long-term caloric restriction and specific deletion of senescence-related T cells are promising interventions to regulate age-related chronic diseases.

## Introduction

Aging might be a modifiable risk factor, and it is feasible to prevent age-related human diseases and frailty by modulating fundamental mechanisms by which aging progresses [1–3]. One such mechanism that has immense potential as a therapeutic target is inflammaging, that is, a smoldering proinflammatory phenotype that accompanies aging in mammals [4, 5]. The

**Funding:** This study was funded by the Japan Society for the Promotion of Science (JSPS KAKENHI Grant Number 22590814) to KS, a research grant from Takeda Medical Research Foundation to KS, and Research Grants in the Natural Sciences from the Mitsubishi Foundation to KS. The funders had no role in study design, data collection and analysis, decision to publish, or preparation of the manuscript.

**Competing interests:** The authors have declared that no competing interests exist.

source of proinflammatory factors underlying inflammaging has been proposed as cells that have acquired a senescence-associated secretory phenotype (SASP) [1–6]. Inflammaging may have multiple causes and is believed to be linked to immunosenescence. Immunosenescence is a multifactorial condition observed in the elderly, consisting of a decreased capacity to respond to vaccination and invading pathogens, and an increased proinflammatory response owing to the accumulation of memory/effector cells [2, 4, 7, 8].

Shimatani *et al*. have demonstrated that age-dependent accumulation of programmed death 1 receptor (PD-1)$^+$ memory-phenotype (MP) T-cell population contributes to the global depression of the T-cell immune response and proinflammatory trait observed in aged mice [9]. This subset of T cells is named "senescence-associated T cells (SA-T cells)", as they hardly proliferate in response to T-cell receptor stimulation and produce an abundance of proinflammatory cytokines, including osteopontin (OPN) at the expense of typical T-cell lymphokines. Although SA-T cells express inhibitory receptors such as PD-1 and cytotoxic T lymphocyte antigen 4, they are distinct from exhausted T cells, and signaling via these receptors does not involve the unique function of these cells [7]. The emergence of SA-T cells is correlated with T-cell homeostatic expansion rather than specific clonal expansion by specific exogenous antigen [8]. Thus, it is possible that SA-T cells play a central role in the progression of immunosenescence in the aged mammals.

Previously, we have reported that a specific subpopulation of CD44$^{high}$ CD62L$^{low}$ CD4$^+$ T cells that express PD-1 and CD153 in high-fat diet (HFD)-induced obese mice [10]. They exhibit senescent phenotypes and cause visceral adipose tissue (VAT) inflammation by producing a large amount of OPN. While putting it together with the report by Shimatani *et al*., SA-T cell subpopulation may be involved in the development of smoldering inflammation associated with both aging and obesity [11, 12]. Although a recent study has demonstrated that the CD153-CpG vaccine successfully reduced the number of SA-T cells in VAT and ameliorated metabolic disorders in HFD-induced obese mice [13], it remains unclear whether the depletion of senescent T cells can retard age-related pathological alterations.

Calorie restriction (CR) extends lifespan and attenuates the onset of age-related diseases in a variety of species [14–18]. CR is also reported to reverse age-related physiological and pathophysiological alterations in mammals. Therefore, it is expected that CR can exert an inhibitory effect against the progression of immunosenescence. The mechanisms underlying CR-mediated preferable effects are not comprehensively elucidated, but weight loss attributed to CR may be the most effective means of improving metabolic disorders, thereby decreasing the risk for the development of insulin sensitivity and/or diabetes mellitus [15, 18]. However, Shirakawa *et al*. have failed to eliminate SA-T cells from VAT by switching from an HFD to a control diet for 8 weeks despite significant weight loss [19]. Therefore, long-term CR may be necessary for the prevention and removal of SA-T cells. The influence of long-term CR on the increase in SA-T cell subpopulation accompanied by aging is yet to be reported. Then, we evaluated the effect of long-term CR on PD-1$^+$ MP CD4$^+$ and CD8$^+$ T cells in the spleen and VAT and glucose tolerance in aged mice. Furthermore, we attempted to identify the effect of immunological depletion of PD-1$^+$ T cells in aged mice.

## Materials and methods

### Mice and ethics approval

We obtained Male C57BL/6J mice from Charles River Japan or Nihon Clea. All mice were housed under a 12-h light-dark cycle and allowed free access to food. This study was conducted in accordance with the Guide for the Care and Use of Laboratory Animals published by the US National Institutes of Health (NIH publication no. 85–23, revised 1996) and was

approved by the Institutional Animal Care and Use Committee at the Keio University School of Medicine and by the Animal Research Committee of Hyogo College of Medicine.

## Caloric restriction protocol

CR was performed as described previously [20, 21]. Fifty-week-old C57BL/6J male mice were housed in individual cages according to the institutional protocols at the Keio University Experimental Animal Centre. Mice were fed ad libitum (AL) with a modified semi-purified diet A (Oriental Yeast Co., Tokyo, Japan) for 2 weeks. The average caloric intake was calculated from the daily food intake during the following 2-week period. After weaning, the mice were randomly divided into two groups. AL mice were continued to be fed AL for 28 weeks using control diet A. CR mice were fed 90% of the average value of caloric intake using a modified semi-purified diet B during the AL period for 2 weeks (10% restriction), followed by 60% of that using a modified semi-purified diet C for 26 weeks (40% restriction). The daily intake of vitamins and minerals was constant throughout the CR period by using semi-purified diets B and C enriched in vitamins and minerals.

## Glucose and insulin tolerance tests

We performed an oral glucose tolerance test (GTT) (oral administration of 1.5 g/kg of glucose, after 16 h fasting) to assess glucose intolerance. We also performed an insulin tolerance test (ITT) (administration of 0.75–2 U/kg of insulin intraperitoneally, after 4 h fasting) to evaluate insulin resistance.

## Measurement of serum insulin levels

Serum insulin, adiponectin and leptin were examined after fasting for 16 h using enzyme-linked immunosorbent assay kits (adiponectin; Sekisui-Kagaku, insulin and leptin; Morinaga).

## Isolation of the stromal vascular fraction and flow cytometry

We isolated stromal vascular cells using previously described methods, with some modifications [10, 19]. The mice were sacrificed under deep anesthesia after systemic heparinization. The epididymal VAT (eVAT) was removed and ground, and incubated for 40 min in collagenase II/DNase I solution (1mg/mLl collagenase II and 50μg/mL in HBSS solution) with gentle stirring. Then, the digested tissue was centrifuged at 1,000 $g$ for 10 min. The resulting pellets were washed twice with cold PBS and filtered through a 70-mm mesh. Red blood cells were lysed with erythrocyte-lysing buffer (eBioscience) for 5 min and resuspended in RPMI-1640 supplemented with 10% FBS.

Single-cell suspensions of splenocytes and an adipose stromal vascular fraction (SVF) were blocked with CD16/32 monoclonal antibody (2.4G2; BD Biosciences) at 4˚C for 5 min. Subsequently, the cells were stained with a mixture of antibodies at 4˚C for 20 min. Flow cytometric analysis and sorting were performed on a FACSAriaIII instrument (BD Biosciences) and performed using FlowJo software (Tree Star). The antibodies used in the present study were specific to CD3 (145-2C11; eBioscience and 17A2; BioLegend), CD4 (GK1.5; eBioscience), CD8a (53–6.7; eBioscience), CD11b (M1/70; eBioscience), CD11c (N418; eBioscience), CD44 (IM7; eBioscience), CD62L (MEL-14; eBioscience), CD206 (MR5D3; BioLegend and C068C2; BioLegend), F4/80 (BM8; eBioscience), PD-1 (J43, RMP1-30; eBioscience and 29F.1A12; BioLegend), P21 (sc-6246, Santa Cruz), and H2AX (N1-431, BD Pharmingen).

### Quantitative analysis of adipocytes and crown-like structures

For adipose tissue, hematoxylin-eosin staining was performed using standard protocols. Images of ten randomly selected high-power fields were acquired (BZ-9000; Keyence, Osaka, Japan), after which the diameters in each field were measured by an observer blinded to the conditions (BZ image analyzer II; Keyence). Furthermore, we counted the numbers of crown-like structures (CLS), defined as an adipocyte with a disrupted plasma membrane surrounded by accumulated cells and/or engulfing macrophages, and other positively stained cells.

### Real-time quantitative PCR

Total RNA samples from adipose tissue were prepared using an RNeasy Mini Kit (Qiagen) or Trizol reagent (Invitrogen), according to the manufacturer's instructions. A First-strand cDNA synthesis kit (Invitrogen) was used for cDNA synthesis. Quantitative real-time PCR was performed using the ABI Prism 7700 sequence detection system (Applied Biosystems) and predesigned gene-specific primer and probe sets (TaqMan Gene Expression Assays, Applied Biosystems). We usec the 18S ribosomal RNA as an internal control.

### Anti-PD-1 antibody treatment

To clarify the essential role of PD-1[+] T cells in adipose tissue inflammation in aged mice, either anti-mouse PD-1 mAb (J43, hamster IgG, Bio-X-Cell) (250 μg/mouse) or control IgG (Bio-X-Cell) was administered intraperitoneally three times per week for three weeks (nine administrations in total) in aged mice fed a normal diet (ND) from the age of 77 weeks. The mice were examined at 80 weeks of age (Fig 4). The anti-PD-1 antibody used in this study (J43) is considered a depleting antibody. J43 triggered complement-dependent cytotoxicity in PD-1[+] T cells *in vitro* and *in vivo* [22].

### Western blotting

Frozen eVAT and liver tissues were homogenized in standard lysis buffer. After centrifugation at 15,000 x *g* for 15 min, the supernatants were collected, and the protein concentration was determined. Equal amounts of total proteins (20–40 μg) were subjected to SDS-PAGE. Antibodies against total Akt and phosphorylated-Akt (at the serine 473 residue) were purchased from Cell Signaling Technology (Beverly, MA, USA). Antibodies against total insulin receptor substrate (IRS) 1 and phosphorylated IRS1 (at the serine 307 residue) were purchased from Santa Cruz Biotechnology (Santa Cruz, CA, USA), and glyceraldehyde 3-phosphate dehydrogenase (GAPDH) was purchased from Millipore (Billerica, MA, USA).

### Statistical analysis

Data values are presented as the mean ± SEM. Comparisons between groups were made using the Mann-Whitney U test, whereas data among multiple groups were compared using either the Kruskal-Wallis test with Dunn's multiple comparison test or Bonferroni post hoc analysis, or two-way ANOVA followed by Tukey's post hoc analysis, as appropriate. A *P*-value of $<0.05$ was considered statistically significant. Statistical analysis was performed using GraphPad Prism 5.0 (GraphPad Prism Software Inc., San Diego, CA, USA) and SPSS 15.0 for Windows (SPSS Inc., Chicago, IL, USA).

## Results

First, we examined the basic characteristics of the aged mice. The body weight and eVAT weight were greater in 100-week-old aged mice than those in 10-week-old young mice

[Fig 1A]. Aged mice exhibited impaired glucose tolerance and insulin resistance (Fig 1B) and showed higher serum levels of insulin and leptin and lower levels of adiponectin (Fig 1C).

Next, we evaluated the characteristic features of SA-T cells in the spleen and eVAT of aged mice. Both PD-1$^+$ CD4$^+$ and CD8$^+$ T cells in aged mice exhibited higher expression of CD44 and lower expression of CD62L (Fig 2A). In addition, the expression of p21 in PD-1$^+$ CD44$^{high}$ T cells was significantly higher than that in PD-1$^-$ CD44$^{high}$ T cells isolated from eVAT (Fig 2B) and the spleen (Fig 2C) of aged mice. Furthermore, we evaluated the expression levels of p16 and H2AX in PD-1+ and PD-1- T cells obtained from the spleen and eVAT. Although there was no difference in the expression levels of p16 (data not shown), we found that the expression levels of H2AX in CD3+ PD-1+ T cells of eVAT were significantly higher than those in CD3+ PD-1- T cells of eVAT, particularly in aged mice (Fig 2D). These results indicate that PD-1$^+$ T cells in aged mice have similar features to PD-1$^+$ CD4$^+$ T cells observed in the eVAT of HFD-induced obese mice.

We examined the distribution of SA-T cells in the spleen and eVAT of mice at the different week-old. The subpopulation of PD-1$^+$ MP (CD44$^{high}$ and CD62L$^{low}$) T cells (both CD4$^+$ and CD8$^+$ T cells) significantly increased in the spleen and eVAT of aged mice (Fig 3A–3D). In particular, the increase in the percentage of PD-1$^+$ MP CD8$^+$ T cells to total CD8+ T cells in eVAT was remarkable in 100-week-old mice (Fig 3D).

Long-term CR substantially decreased body weight and eVAT weight when compared with age-matched mice fed AL (Fig 4A and 4B). CR improved glucose tolerance and insulin resistance (Fig 3C), lowered baseline serum insulin and leptin levels, and increased serum adiponectin levels (Fig 4D).

Although the percentage of PD-1$^+$ MP CD4$^+$ T cells to total MP CD4$^+$ T cells was similar between AL-fed and CR-treated mice, the number of PD-1$^+$ MP CD4$^+$ T cells in eVAT per gram of adipose tissue was markedly lower in CR mice than in age-matched mice (Fig 5A and 5B). In contrast, long-term CR significantly reduced the percentage and the number of PD-1$^+$

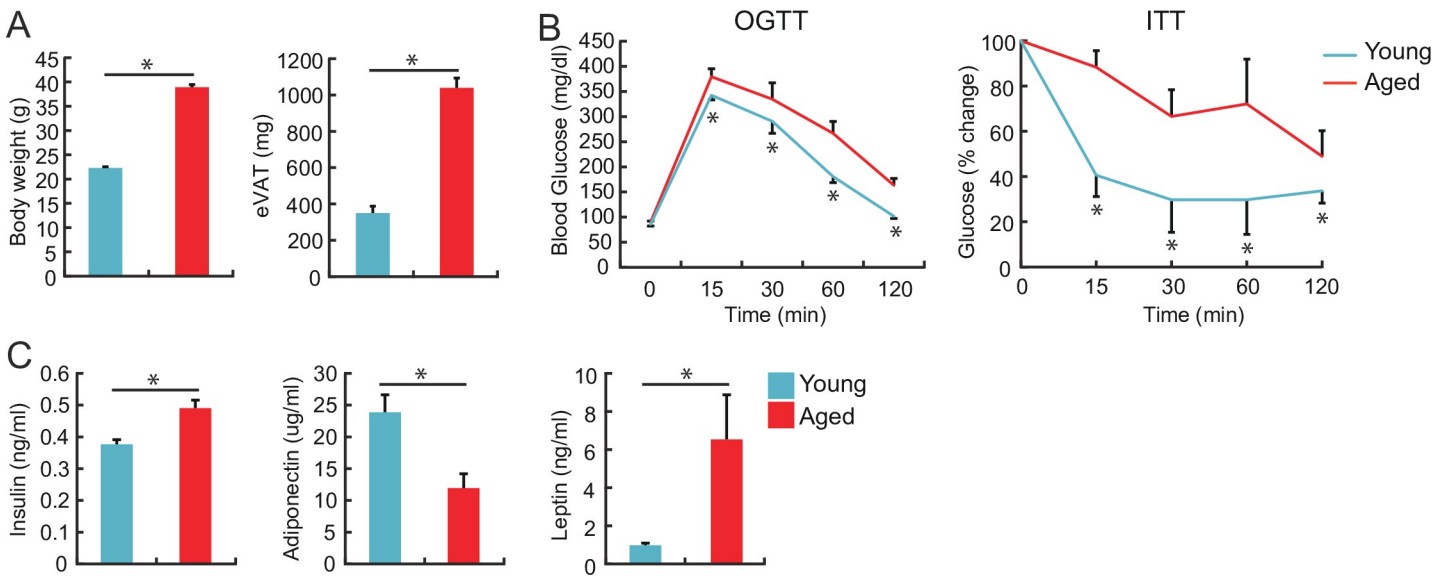

**Fig 1. Increase in body and visceral adipose tissue (VAT) weight and the development of insulin resistance with advanced age.** The body and epididymal VAT (eVAT) weights (**A**), results of oral glucose tolerance test (OGTT), and insulin tolerance test (ITT) (**B**) were compared between young and aged wild-type (WT) mice fed normal diet (ND) (10-week-old and 100-week-old, respectively) ($n = 6$–$8$ animals in each group). (**C**) Serum levels of insulin, adiponectin, and leptin were compared between young and aged WT mice fed a normal diet (ND) (10-week-old and 100-week-old, respectively) (n = 6–8 animals in each group). $^*P < 0.05$; NS: not significant. Data are represented as the mean ± SEM.

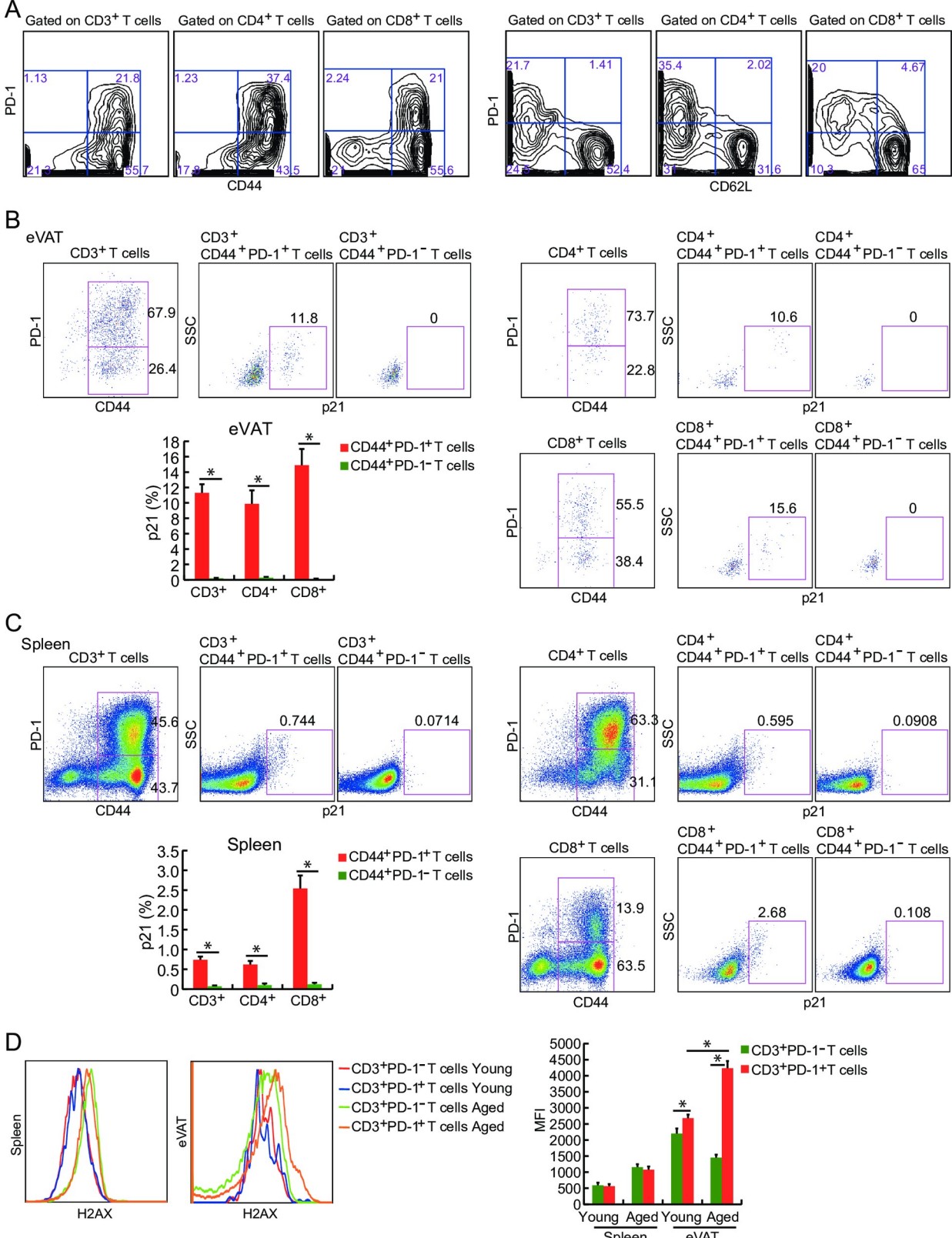

**Fig 2. PD-1⁺ T cells observed in aged mice are memory-phenotype (MP) and express senescence marker, p21.** (**A**) Spleen cells from 80-week-old mice were stained for CD3, CD4, CD8, CD44, and CD62L, and then analyzed by flow cytometry. Results indicated that PD-1⁺ T cells are

mainly MP because they expressed higher levels of CD44 and lower levels of CD62L. (**B, C**) Flow cytometric analysis of p21 expression in PD-1+ CD44+ CD4+ T cells and PD-1- CD44+ CD4+ T cells obtained from eVAT and the spleen of 80-week-old mice (*n* = 5–6 mice in each group). (**D**) Flow cytometric analysis of H2AX expression in PD-1+ CD3+ T cells and PD-1- CD3+ T cells obtained from eVAT and the spleen of either 11-week-old or 80-week-old mice (*n* = 5–6 mice in each group). *P* < 0.05; NS: not significant. Data are represented as the mean ± SEM.

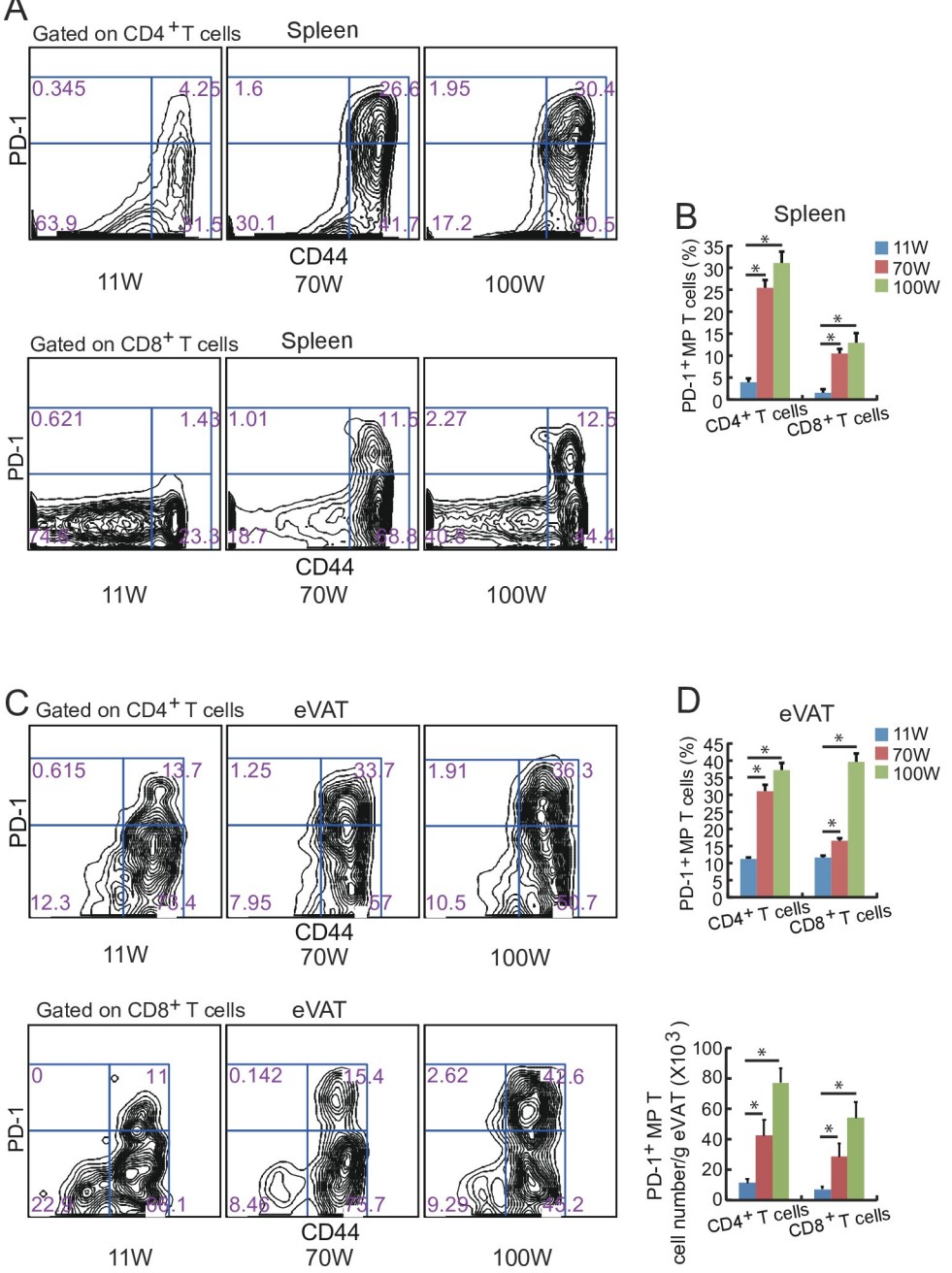

**Fig 3. The increase in the PD-1+ T-cell subpopulation is associated with advanced age.** (**A, B**) Flow cytometric analysis of PD-1+ CD44+ CD4+ T cells and PD-1+ CD44+ CD8+ T cells in spleen cells of WT mice of different ages fed with a ND (*n* = 5–6 mice in each group). (**C, D**) Flow cytometric analysis of PD-1+ CD44+ CD4+ T cells and PD-1+ CD44+ CD8+ T cells from the stromal vascular fraction (SVF) from the eVAT of WT mice of different age fed an ND. (*n* = 5–6 mice in each group). *P* < 0.05; NS: not significant. Data are represented as the mean ± SEM.

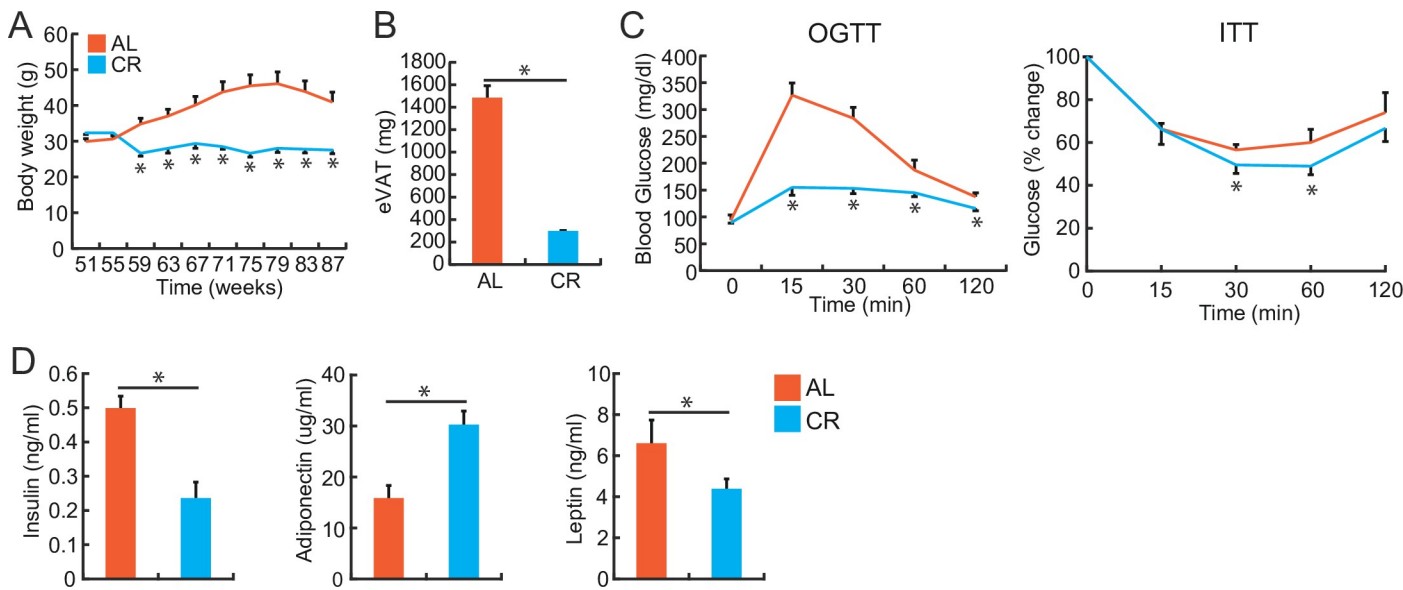

**Fig 4. Caloric restriction (CR) improves glucose intolerance and insulin sensitivity during aging.** Fifty-two-week-old mice were fed either ad libitum (AL) or treated with caloric restriction (CR) and analyzed on reaching 80 weeks of age. (**A**) Changes in body weight during the time course. (**B**) eVAT weights of the AL and CR mice. (**C**) Results of OGTT and ITT in WT mice fed AL or CR. (**D**) Serum levels of insulin, adiponectin, and leptin in the AL and CR mice. *$P < 0.05$. Data are represented as the mean ± SEM.

MP CD8+ T cells in the eVAT. Notably, CR also reduced the percentage of PD-1+ MP CD4+ and CD8+ T cells in the spleen of aged mice (Fig 5C and 5D).

In the eVAT of AL-fed aged mice, the distribution of macrophage subtypes exhibited an M2-type predominant pattern (Fig 5E and 5F). Furthermore, long-term CR significantly decreased the total number of macrophages in eVAT. In particular, CR decreased the number of M1 type (CD11b+) macrophages per gram of adipose tissue and further shifted to a pattern of M2-type predominant (Fig 5E and 5F). Accordingly, CR decreased the average diameter of adipocytes and the number of CLS (Fig 5G and 5H). Finally, CR significantly decreased the expression levels of tumor necrosis factor (TNF)-α, interleukin (IL)-6, chemokine ligand (CCL)2, and OPN in the eVAT of aged mice (Fig 5I).

Next, we examined whether ablation of adipose PD-1+ T cells reduced adipose inflammation and insulin resistance in aged mice. We compared food intake, BW, OGTT, and ITT before and after either control IgG or anti-PD-1 antibody treatment. PD-1+ cell-depleting antibody treatment significantly improved glucose tolerance and attenuated insulin resistance without affecting food intake, body weight, or eVAT weight (Fig 6A–6E). Furthermore, the treatment with PD-1+ cell-depleting antibodies decreased serum insulin and leptin levels and increased serum adiponectin levels (Fig 6D). To examine the effect of anti-PD-1 antibody treatment on intracellular insulin signaling, we evaluated the phosphorylation levels of Akt/PKB at the serine 473 residue and those of IRS1 at the serine 307 residue. Phosphorylation levels of IRS1 at baseline were higher in livers obtained from aged mice treated with control IgG than in those from aged mice treated with anti-PD-1 antibody (Fig 3G). In contrast, phosphorylation levels of Akt at baseline were lower in both liver and eVAT obtained from aged mice treated with control IgG, compared with those from aged mice treated with anti-PD-1 antibody (Fig 6G and 6H). In addition, insulin-induced phosphorylation of Akt was significantly improved in eVAT obtained from aged mice treated with anti-PD-1 antibody (Fig 6H).

The treatment with PD-1+ cell-depleting antibodies decreased the percentage of CD4+ and CD8+ MP T cells, paralleling the increased percentage of naïve T cells in both the spleen and

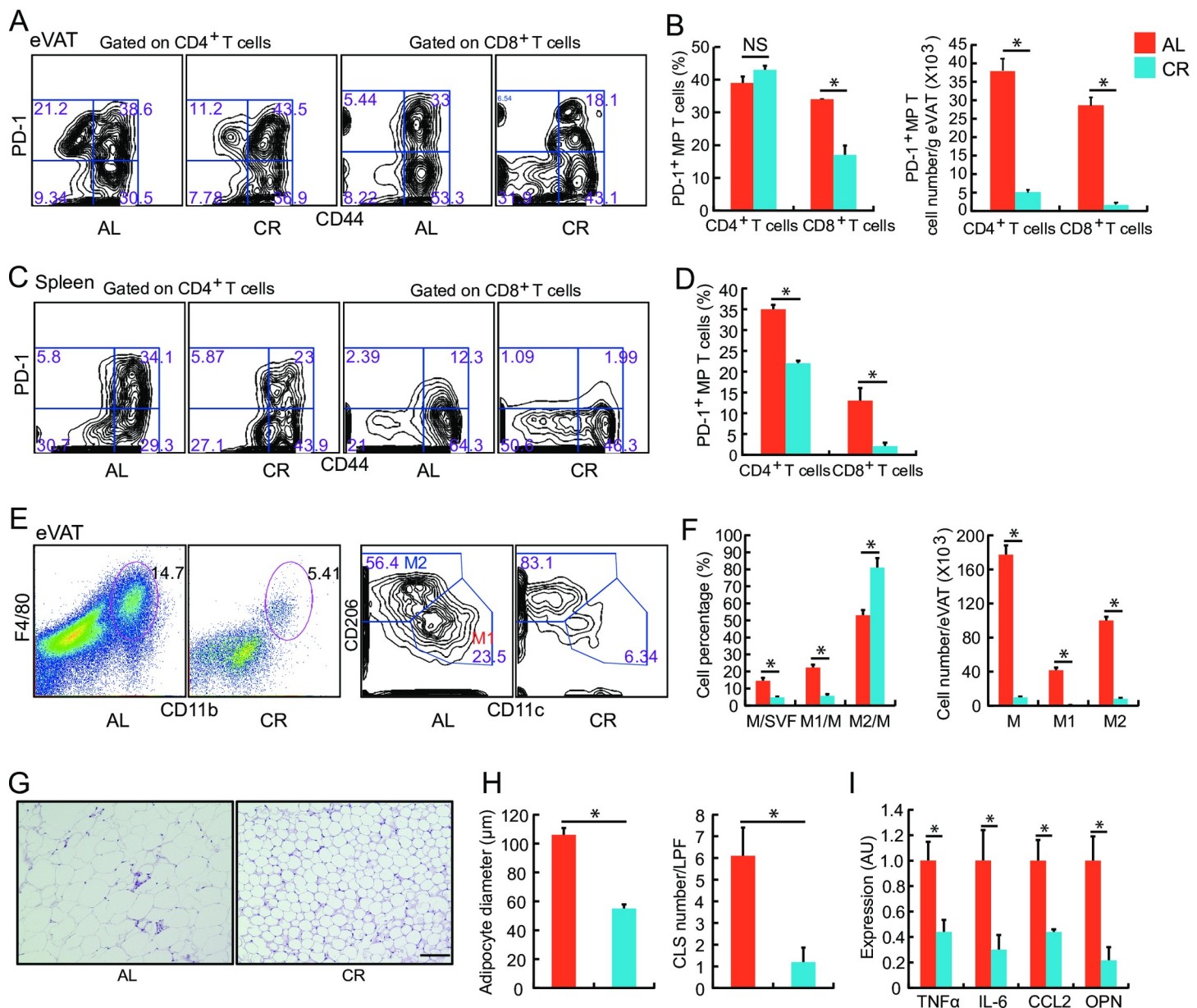

**Fig 5. Effects of caloric restriction (CR) on the PD-1⁺ T-cell subpopulation and adipose inflammation in aged mice.** Fifty-two-week-old mice were fed either ad libitum (AL) or treated with caloric restriction (CR) and analyzed on reaching 80 weeks of age. (**A, B**) Flow cytometric analysis of PD-1⁺ CD44⁺ T cells (PD-1⁺ MP T cells) in SVF from eVAT of the AL and CR mice. (**C, D**) Flow cytometric analysis of PD-1⁺ CD44⁺ T cells (PD-1⁺ MP T cells) was performed using spleen cells of the AL and CR mice. (**E, F**) Flow cytometric analysis of M1 type macrophages (CD11b⁺ F4/80⁺ CD11c⁺ CD206⁻) and M2 type macrophages (CD11b⁺ F4/80⁺ CD11c⁻ CD206⁺) in the eVAT of AL and CR mice. (**G**) Representative images of H&E staining of eVAT from AL and CR mice. Scale bar, 100μm. (**H**) Mean adipocyte diameter and the number of crown-like structures (CLS) in eVAT were determined. (**I**) Real-time PCR analysis of cytokine expression in eVAT. The expression levels of each transcript were normalized to those in the mice fed AL (n = 7–9 mice in each group). *P < 0.05; NS: not significant. Data are represented as the mean ± SEM. SVF; stromal vascular fraction, TNF-α; tumor necrosis factor-α, IL-6; interleukin-6, CCL2; chemokine ligand 2, OPN; osteopontin. AU: arbitrary units.

eVAT of aged mice (Fig 7A and 7B). We confirmed the effective removal of PD-1+ T cells, especially PD-1+ CD4+ T cells, from the eVAT and spleen by anti-PD-1 antibody treatment (Fig 7C–7F). Additionally, the treatment with PD-1⁺ cell-depleting antibodies decreased the percentage of M1 type macrophages without affecting the percentage of M2 type macrophages (Fig 7G). Furthermore, the treatment with PD-1⁺ cell-depleting antibodies decreased the

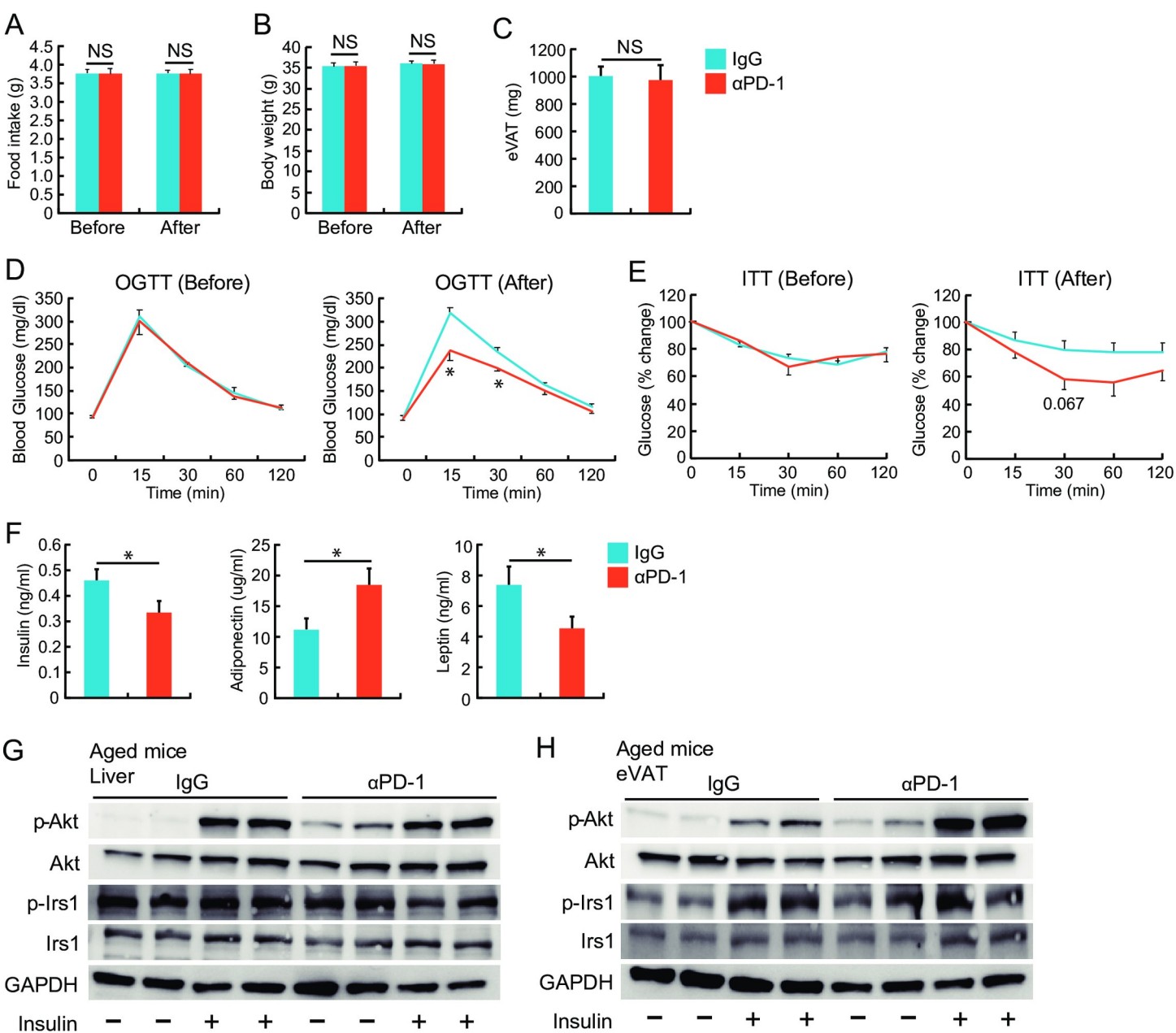

**Fig 6. Effects of anti-PD-1 antibody treatment on insulin resistance in aged mice.** Either anti-mouse PD-1 antibody (αPD-1) or control IgG (IgG) was administered three times per week for three weeks to 77-week-old mice. All examinations were performed when mice reached 80 weeks of age (*n* = 6–8 mice in each group). (**A**) Daily food intake and (**B**) body weight (BW) were compared between IgG-treated and αPD-1-treated aged mice before and after treatment. (**C**) eVAT weight was compared between aged mice treated with either αPD-1 or IgG. (**D, E**) Results of the OGTT and ITT between IgG-treated and αPD-1-treated aged mice before and after treatment, respectively. (**F**) Serum levels of insulin, adiponectin, and leptin in aged mice treated with αPD-1 or IgG. Representative Western blot images showing total and phosphorylated Akt/PKB at the serine 473 residue, total and phosphorylated insulin receptor substrate (IRS) 1 at the serine 307 residue, and GAPDH expression levels, at baseline and after insulin administration in the liver (**G**) and eVAT (**H**) obtained from aged mice treated with either IgG or αPD-1. *$P < 0.05$; NS: not significant. Data are represented as the mean ± SEM.

number of CLS without affecting the average diameter of adipocytes (Fig 7H). Finally, the treatment with PD-1[+] cell-depleting antibodies decreased the expression levels of proinflammatory cytokines in eVAT, except for TNF-α (Fig 7I).

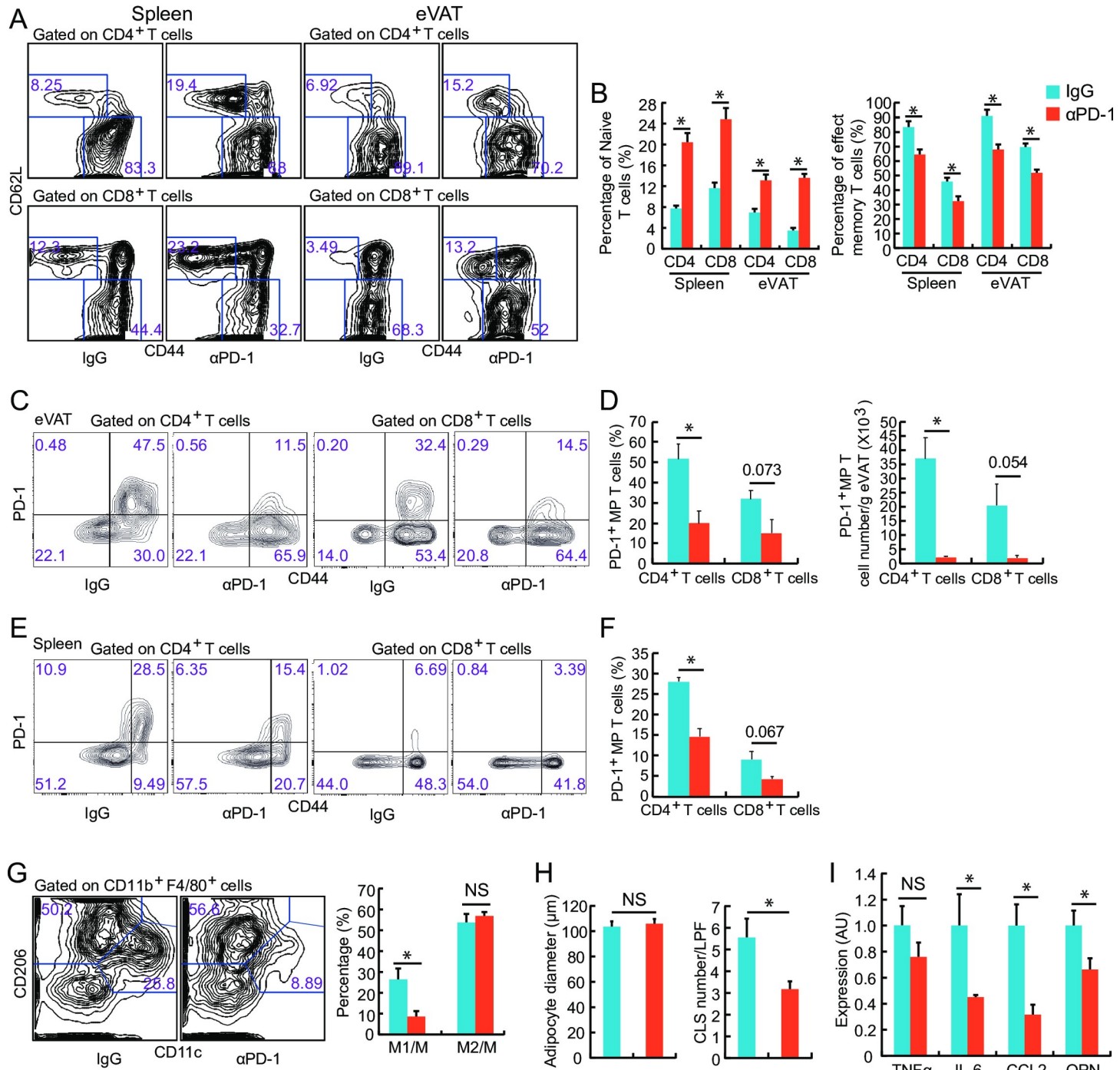

**Fig 7. Effects of anti-PD-1 antibody treatment on adipose inflammation in aged mice.** Either anti-mouse PD-1 antibody (αPD-1) or control IgG (IgG) was administered three times per week for three weeks to 77-week-old mice. All examinations were performed when mice reached 80 weeks of age ($n$ = 4–8 mice in each group). (**A**) Flow cytometric analysis of CD44⁻ CD62L⁺ T cells (naïve T cells) and CD44⁺ CD62L⁻ T cells (MP T cells) was performed using spleen cells or eVAT of AL and CR mice. (**B**) The percentages of CD44⁻ CD62L⁺ T cells and CD44⁺ CD62L⁻ T cells. (**C, D**) Flow cytometric analysis of PD-1⁺ CD44⁺ T cells (PD-1⁺ MP T cells) in SVF from eVAT from aged mice treated with either αPD-1 or IgG. (**E, F**) Flow cytometric analysis of PD-1⁺ CD44⁺ T cells (PD-1⁺ MP T cells) was performed using spleen cells from aged mice treated with either IgG or αPD-1. (**G**) Flow cytometric analysis of M1 type macrophages (CD11b⁺ F4/80⁺ CD11c⁺ CD206⁻) and M2 type macrophages (CD11b⁺ F4/80⁺ CD11c⁻ CD206⁺) in eVAT from aged mice treated with either αPD-1 or IgG. (**H**) Mean adipocyte diameter and the number of CLS in eVAT. (**I**) Real-time PCR analysis of cytokine expression in eVAT. The expression levels of each transcript were normalized to those in aged mice treated with IgG ($n$ = 6–8 mice in each group). AU: arbitrary units. $^{*}P$ < 0.05; NS: not significant. Data are represented as the mean ± SEM.

## Discussion

We demonstrated that the SA-T cell subpopulation increased with advanced age in the spleen, and age-related adiposity was associated with an accumulation of SA-T cells in eVAT. Long-term CR prevented this age-related adiposity and improved insulin resistance in aged mice. Furthermore, selective reduction of PD-1+ T cells by specific antibody treatment attenuated adipose inflammation and improved insulin resistance in aged mice. Taken together with our previous report [10], we concluded that the emergence of SA-T cells is a common molecular pathway in the development and progression of VAT inflammation and insulin resistance in the context of both chronological aging and obesity.

The mechanism by which SA-T cells increase with advancing age remains unknown. We previously found that antigen-presenting cells, including macrophages and B cells, are necessary to induce SA-T cells in HFD-induced obese mice [10]. This indicates that SA-T cells are derived from self-reactive T cells in the eVAT. In addition, PD-1+ T cells increased with advanced age and exhibited higher expression levels of p21 and H2AX (Figs 1 and 2). These results are consistent with the findings observed in PD-1+ CD4+ T cells obtained from the eVAT of HFD-induced obese mice [10]. Therefore, we speculate that the appearance of PD-1 + T cells is closely associated with cellular senescence induced DNA damage. We speculate that the suppression of oxidative stress that causes DNA damage contributes, at least in part, to the reduction of T-cell immunosenescence in aged mice treated with CR because long-term CR attenuates oxidative damage accumulated with advanced age in various organs [16–18]. In addition, the modification of p21/p53 signaling in T cells may have the potential to suppress T-cell immunosenescence.

Systemic insulin resistance develops in not only obese individuals, but also aged individuals. Chronic low-grade inflammation in eVAT plays a role in the progression of insulin resistance in these individuals [12, 23, 24]. Increasing evidence demonstrates that insulin resistance promotes the aging process, and, in reverse, aged individuals become vulnerable to insulin resistance [11, 24]. Thus, we supposed the close relationship between obesity and aging and hypothesized that they might be tied by SA-T cells. In the present study, anti-PD-1 antibody treatment effectively removed SA-T cells and improved insulin signaling, especially in eVAT (Figs 6 and 7). Thus, we believe that reduced SA-T cell subpopulation in eVAT by long-term CR contributes to the improvement of insulin resistance in aged mice. However, the effect of long-term CR on insulin resistance is multifaceted and might be mediated via mixed mechanisms, including the effect on glucose metabolism in the liver and skeletal muscle [18, 25–27]. In addition, the maintenance of adequate adiposity by CR must be important to keep healthy condition during the advanced age period. Because age-associated changes in adiposity loses relevant secretion of adipocytokines, leading to chronic low-grade inflammation.

While searching the literatures, the effect of CR on immunosenescence remains controversial [28–33]. CR reportedly delayed T-cell senescence in nonhuman primates [30]. In particular, long-term CR maintains a higher percentage of both CD4+ and CD8+ T naïve cells in peripheral blood mononuclear cells, spleen, and other organs of CR-treated aged animals [28–30, 32, 33]. Furthermore, long-term CR preserved T cell receptor repertoire diversity, maintained T-cell proliferation ability, and decreased the production of proinflammatory cytokines such as interferon γ [30]. In contrast, CR is reported to exaggerate age-associated changes in Natural Killer cell phenotype [32]. Accordingly, the negative effect of CR on some kinds of viral infection susceptibility is becoming clear [28]. Another report demonstrated that the preferable effects of CR on T-cell immunosenescence were lost by either early or late CR onset [29]. In addition, the impact of CR on lymphocyte maturation and senescence seems different

between mouse strains [32]. In the present study, we found that CR decrease the emergence of SA-T cells in both the spleen and eVAT associated with advanced age, suggesting that the effect of CR on T-cell senescence is systemic in CR-treated aged mice. In this regard, White *et al.* reported that the effect of CR on lymphoid cell population in spleen were identical to those observed in lung, liver, and lymph nodes [32]. Trott *et al.* evaluated the effect of CR on the immune cell infiltration in the aorta and mesenteric arteries including surrounding perivascular adipose tissue [31]. Age-related increases in CD4$^+$ and CD8$^+$ T cell, B cell, and macrophage infiltration in the vascular system were attenuated by CR in aged mice. Taken together, we could not conclude that CR is an optimal strategy for maintaining healthy immune function, but speculated that CR and CR mimetics, that can reproduce the preferable effects of CR, may have a therapeutic potential for retarding the progression of inflammaging via a SASP-dependent mechanism. Indeed, both long-term CR and specific antibody treatment increased the percentage of naïve CD4$^+$ T cells in aged animals [30].

Increasing evidence demonstrates that the depletion of senescent cells from the whole body, called senolysis, might increase lifespan and attenuate age-related chronic diseases [1, 3, 34, 35]. Thus, SA-T cells are potential therapeutic targets for senolysis [8, 10]. In this regard, Yoshida *et al.* have clearly demonstrated that the CD153-CpG vaccine is promising for a senotherapeutic method for preventing the accumulation of SA-T cells in mice [13]. However, it remains unclear whether the depletion of SA-T cells influences age-related physiological and pathophysiological alterations. In this study, we revealed that both long-term CR and immunological intervention against PD-1 attenuated adipose inflammation and improved insulin resistance in aged mice. The anti-PD-1 antibody used in this study (J43) is considered a depleting antibody. J43 triggered complement-dependent cytotoxicity in PD-1$^+$ T cells *in vitro* and *in vivo* [22]. In fact, treatment with J43 antibody effectively reduced the number of PD-1$^+$ CD4$^+$ T cells in both the eVAT and spleen (Fig 7C–7F). Therefore, we speculate that the depletion of SA-T cells by J43 antibody treatment is the most relevant to the attenuation of adipose inflammation and the improvement in insulin resistance by reducing proinflammatory cytokines.

This study has several limitations as the number of aged and long-term CR-treated mice was limited at the time of this study. First, the characteristic analysis of SA-T cells in aged mice was inadequate. We speculated that SA-T cells observed in the eVAT of aged mice were similar to those in eVAT of HFD-induced obese mice because they exhibited memory-phenotype and expressed higher levels of p21 and H2AX as well as PD-1$^+$ CD4$^+$ T cells in eVAT of HFD-induced obese mice did [10]. However, we did not compare genetic profiles between two PD-1$^+$ MP T-cell subpopulations. Therefore, we cannot conclude that SA-T cells observed in aged mice are identical to those in the eVAT of HFD-induced obese mice. Second, the physiological and pathophysiological role of increased SA-T cells in aging remains unclear. We did not measure SA-T cells in other tissues such as the thymus and lymph nodes. We focused on the impact of T-cell immunesenescence on glucose intolerance associated with age. Therefore, we evaluated the increase in SA-T cells in eVAT and the spleen in mice of different ages because we hypothesized that the T-cell immunosenescence in the spleen and eVAT was closely related to chronic inflammation in aged mice. However, T-cell immunosenescence must be observed and may play a role in other tissues of aged animals. We would like to evaluate SA-T cells in other tissues in a future study. Finally, we did not evaluate the effect of anti-CD153 antibodies treatment in aged mice because specific antibodies against CD153 for the removal of CD153$^+$ PD-1$^+$ T cells were not available. If a unique population of CD153$^+$ PD-1$^+$ T cells plays a central role in the development of SASP-related adipose inflammation [10], the use of specific antibodies against CD153 might be more effective to suppress smoldering inflammation.

## Conclusions

We confirmed that aging increases the specific MP T-cell subpopulation expressing PD-1 in both the spleen and eVAT. Long-term CR prevented age-related adiposity and improved insulin resistance in aged mice. Treatment with specific antibodies against PD-1 reduced adipose inflammation and improved insulin resistance in aged mice. We speculated that both long-term CR and treatment with specific antibodies against PD-1 could prevent smoldering inflammation associated with aging by breaking down a detrimental loop of SASP. Thus, specific targeting of SA-T cells is not only a novel therapeutic strategy for managing obesity-related metabolic disorders but may be a promising intervention to control age-related chronic diseases, as well as the aging process itself.

## Supporting information

**S1 Fig. Gating strategy of immune cell populations in the stromal vascular fraction (SVF).** Representative flow cytometric analysis of PD-1$^+$ CD44$^+$ T cells (PD-1$^+$ MP T cells) from the spleen (**A**) and eVAT (**B**). (**C**) Representative flow cytometric analysis of CD11b$^+$ F4/80$^+$ macrophages; M1 type (CD11b$^+$ F4/80$^+$ CD11c$^+$ CD206$^-$) and M2 type (CD11b$^+$ F4/80$^+$ CD11c$^-$ CD206$^+$) macrophages from eVAT.
(DOCX)

**S1 File. Minimal data set.**
(DOCX)

## Acknowledgments

We would like to thank Dr. Hideo Yagita (Department of Immunology, Juntendo University School of Medicine) for permitting the use of the specific antibody to PD-1 (J43) in this study.

## Author Contributions

**Conceptualization:** Xiaoxiang Yan, Motoaki Sano, Ken Shinmura.

**Formal analysis:** Xiaoxiang Yan, Motoaki Sano, Ken Shinmura.

**Funding acquisition:** Ken Shinmura.

**Investigation:** Xiaoxiang Yan, Natsumi Imano, Kayoko Tamaki, Ken Shinmura.

**Methodology:** Xiaoxiang Yan, Motoaki Sano, Ken Shinmura.

**Writing – original draft:** Ken Shinmura.

**Writing – review & editing:** Xiaoxiang Yan, Motoaki Sano, Ken Shinmura.

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
