## [Decision Letter · Decision Letter 0]

18 Feb 2021

PONE-D-21-02437

The effect of caloric restriction on the increase in senescence-associated T cells and metabolic disorders in aged mice

PLOS ONE

Dear Dr. Shinmura,

Thank you for submitting your manuscript to PLOS ONE. After careful consideration, we feel that it has merit but does not fully meet PLOS ONE’s publication criteria as it currently stands. Therefore, we invite you to submit a revised version of the manuscript that addresses the points raised during the review process.

We look forward to receiving your revised manuscript.

Kind regards,

Ichiro Manabe

Academic Editor

PLOS ONE

Journal Requirements:

2.Thank you for stating the following in the Acknowledgments Section of your manuscript:

"This study was supported in part by JSPS KAKENHI Grant Number 22590814 (2013-2015), Takeda Medical Research Foundation ID201409 (2014), and Research Grants in the Natural Sciences form the Mitsubishi Foundation ID29122 (2017) (to Dr. Shinmura)."

 "All authors declare no financial disclosure. The funders had no role in study design, data collection and analysis, decision to publish, or preparation of the manuscript."

Reviewers' comments:

Reviewer's Responses to Questions

**Comments to the Author**

1. Is the manuscript technically sound, and do the data support the conclusions?

Reviewer #1: Yes

Reviewer #2: Yes

2. Has the statistical analysis been performed appropriately and rigorously? 

Reviewer #1: Yes

Reviewer #2: Yes

3. Have the authors made all data underlying the findings in their manuscript fully available?

Reviewer #1: Yes

Reviewer #2: Yes

4. Is the manuscript presented in an intelligible fashion and written in standard English?

Reviewer #1: Yes

Reviewer #2: Yes

5. Review Comments to the Author

Reviewer #1: In general, this research is a good piece of work to expand the knowledge of CR science. The authors showed that SA-T cells are responsible for the development of age-associated metabolic disorders and long-term CR could decrease the number of SA-T cells in mice spleen and adipose tissue. Furthermore, the immunological depletion of PD-1+ T cells improved insulin resistance in aged mice with the reduction of adipose tissue inflammation. Thus, the authors conclude that SA-T cells are attractive therapeutic target for the treatment of age-related chronic inflammation and insulin resistance.

Major point

To clarify the mechanism of increased insulin sensitivity in aPD-1 treated eWAT, the authors should measure the intracellular insulin signaling molecules and their receptors activation status in the liver and adipose tissue. Furthermore, the measurement of another senescent tissue protein marker would help to understand the pathophysiological roles of PD-1+ cell depletion. At lease, the authors should describe and discuss more these points. (Fig. 6)

Also, the authors should discuss the reason why the PD-1＋ T cells increase with age.

Minor points

p4, line 96

The authors should clarify why they did not measure PD-1+ MP CD4+ and CD8+ cells in the other tissues such as thymus.

Also, the authors should clarify that the PD-1+ MP T cells have low level of CD62L or not. (Fig. 5 and 7)

p5, line 161, p9 line 283, p10 line 329, Fig. 5, and Fig. 7.

Typo?

Crown-like structures should be abbreviated CLS.

Reviewer #2: The authors evaluated the accumulation of SA-T cells in the visceral adipose tissue and spleen under the caloric restriction (CR) in aged mice. Long-term CR decreased the number of PD-1-positive CD4 or CD8 T cells and decreased M1 macrophage. In addition, the immunological depletion of PD-1 T cells reduced adipose inflammation and improved insulin resistance in aged mice. This study includes an interesting finding and the experiment is well designed. Our concerns are as follows.

1. In Figure 2, the percentage of p21 in eVAT is much higher than that in spleen. Could you evaluate the SA-T cells in eVAT or spleenusing other senescence marker, such as DNA damage marker?

2. In Figure 3 and 5, could you also show the total number of CD4 and CD8 T cells in aged mice or under CR condition? The reviewer also would like to know the percentage of Naïve T cells under CR condition.

3. In Figure 6 and 7, could you show the comparison of each parameter before and after PD-1 depletion antibody? These additional experiments will teach us the potential (prevention or improvement) of senolytic therapy.

6. PLOS authors have the option to publish the peer review history of their article (what does this mean?). If published, this will include your full peer review and any attached files.

Reviewer #1: No

Reviewer #2: **Yes: **Hironori Nakagami

---

## [Author Response · Author response to Decision Letter 0]

7 May 2021

Responses to reviewer #1

We would like to thank the reviewer for his/her constructive criticism and helpful suggestions, which have helped us to revise and improve our manuscript. We would also like to thank the reviewer for his/her valuable remarks.

In general, this research is a good piece of work to expand the knowledge of CR science. The authors showed that SA-T cells are responsible for the development of age-associated metabolic disorders and long-term CR could decrease the number of SA-T cells in mice spleen and adipose tissue. Furthermore, the immunological depletion of PD-1+ T cells improved insulin resistance in aged mice with the reduction of adipose tissue inflammation. Thus, the authors conclude that SA-T cells are attractive therapeutic target for the treatment of age-related chronic inflammation and insulin resistance..

#1. To clarify the mechanism of increased insulin sensitivity in aPD-1 treated eWAT, the authors should measure the intracellular insulin signaling molecules and their receptors activation status in the liver and adipose tissue.

We would like to thank you for your constructive criticism and helpful suggestions, which have helped us revise and improve our manuscript. To examine the effect of anti-PD-1 antibody treatment on intracellular insulin signaling, we evaluated phosphorylation levels of Akt/PKB at the serine 473 residue and those of insulin receptor substrate (IRS) 1 at the serine 307 residue. Phosphorylation levels of IRS1 at baseline were higher in livers obtained from aged mice treated with control IgG than in those from aged mice treated with anti-PD-1 antibody (Fig. 6G). In contrast, phosphorylation levels of Akt at baseline were lower in both the liver and eVAT obtained from aged mice treated with control IgG, compared with those from aged mice treated with anti-PD-1 antibody (Fig. 6G and 6H). In addition, insulin-induced phosphorylation of Akt was significantly improved in eVAT obtained from aged mice treated with anti-PD-1 antibody (Fig. 6H). These results strongly suggest that anti-PD-1 antibody treatment improved insulin signaling, especially in eVAT, by reducing proinflammatory cytokines that could induce insulin resistance. Accordingly, we added each paragraph in the Methods and Results section, and Fig. 6 legend of the revised manuscript (P6, L184-L193; P10, L322-L331; and P10, L344-L348, respectively).

#2. Furthermore, the measurement of another senescent tissue protein marker would help to understand the pathophysiological roles of PD-1+ cell depletion. At lease, the authors should describe and discuss more these points. (Fig. 6) 

In addition to p21, we evaluated the expression levels of p16 and H2AX between PD-1+ and PD-1- CD3+ T cells obtained from the spleen and eVAT. Although there was no difference in the expression levels of p16, we found that the expression levels of H2AX in PD-1+ CD3+ T cells of eVAT were significantly higher than those in PD-1- CD3+ T cells obtained from eVAT, particularly in aged mice (Fig. 2D). Thus, PD-1+ T cells increase with aging and exhibit higher expression levels of p21 and H2AX. These results are consistent with the findings observed in PD-1+ CD4+ T cells obtained from eVAT of high-fat diet-induced obese mice (Shirakawa et al. J Clin Invest 2016;126:4626-39.). Therefore, we speculate that the appearance of PD-1+ T cells in eVAT is closely associated with cellular senescence induced DNA damage. In addition to PD-1+ T cell depletion, the modification of p21/p53 signaling in T cells might have the potential to suppress T-cell immunosenescence. Any interventions for the purpose of reducing DNA damage, such as the suppression of oxidative stress in the circulating environment, might become a different approach to reduce T-cell immunosenescence. Therefore, we added the findings of H2AX analysis in the revised Fig. 2D (Fig. 2 legend, P8, L242-L246) and the results (P7, L228-L232), and this assessment in the Discussion (P12, L394-L407).

#3. Also, the authors should discuss the reason why the PD-1＋ T cells increase with age.

We discussed this issue as described above and inserted a paragraph in the Discussion section (P12, L394-L407).

#4. p4, line 96

The authors should clarify why they did not measure PD-1+ MP CD4+ and CD8+ cells in the other tissues such as thymus. 

Thank you very much for your constructive advice. In the present study, we focused on the impact of T-cell immunesenescence on glucose intolerance associated with aging. Therefore, we planned to evaluate the increase in PD-1+ T cells in eVAT and the spleen in mice of different ages because we supposed that the T-cell subpopulation in the spleen and eVAT was closely related to chronic inflammation in aged mice. As the reviewer pointed out, T-cell immunesenescence must be observed and might play a role in the thymus and lymph nodes. We would like to evaluate PD-1+ T cells in these tissues in a future study. We briefly describe this as a limitation of the present study (P14, L477-L484).

#5. Also, the authors should clarify that the PD-1+ MP T cells have low level of CD62L or not. (Fig. 5 and 7)

The PD-1+ T cells exhibited lower levels of CD62L (Fig. 2A). Thus, PD-1+ T cells are classified as memory phenotypes, which are characterized by higher expression of CD44, and lower expression of CD62L. Unfortunately, the number of immune cells obtained from eVAT is relatively low in CR mice and aged mice treated with anti-PD-1 antibody and we divided these immune cells into three tubes for the analysis of macrophages, CD3+ T cells, and regulatory T cells (Treg) (in the present study, we did not show the results of Treg analysis). Therefore, we could not evaluate the expression levels of CD62L in PD-1+ CD44high CD3+ T cells by FACS in CR mice and aged mice treated with anti-PD-1 antibody. However, we believe that it is reasonable to expect that PD-1+ T cells obtained from CR mice and aged mice treated with anti-PD-1 antibody are the same as aged mice (Fig. 2A).

#6. p5, line 161, p9 line 283, p10 line 329, Fig. 5, and Fig. 7.

Typo?

Crown-like structures should be abbreviated CLS. 

Thank you very much for identifying this issue. We have corrected all the typing errors.

Thank you again for your excellent suggestions.

 

Response to reviewer #2

We would like to thank the reviewer for his/her constructive criticism and helpful suggestions, which have helped us to revise and improve our manuscript, as well as for his/her valuable remarks.

The authors evaluated the accumulation of SA-T cells in the visceral adipose tissue and spleen under the caloric restriction (CR) in aged mice. Long-term CR decreased the number of PD-1-positive CD4 or CD8 T cells and decreased M1 macrophage. In addition, the immunological depletion of PD-1 T cells reduced adipose inflammation and improved insulin resistance in aged mice. This study includes an interesting finding and the experiment is well designed. Our concerns are as follows.

#1. In Figure 2, the percentage of p21 in eVAT is much higher than that in spleen. Could you evaluate the SA-T cells in eVAT or spleen using other senescence marker, such as DNA damage marker?

In addition to p21, we evaluated the expression levels of p16 and H2AX between PD-1+ and PD-1- CD3+ T cells obtained from the spleen and eVAT. Although there was no difference in the expression levels of p16, we found that the expression levels of H2AX in PD-1+ CD3+ T cells of eVAT were significantly higher than those in PD-1- CD3+ T cells of eVAT, particularly in aged mice (Fig. 2D). Thus, PD-1+ T cells increase with aging and exhibit higher expression levels of p21 and H2AX. These results are consistent with the findings observed in PD-1+ CD4+ T cells obtained from eVAT of high-fat diet-induced obese mice (Shirakawa et al. J Clin Invest 2016;126:4626-39.). Therefore, we speculate that the appearance of PD-1+ T cells in eVAT is closely associated with cellular senescence induced DNA damage. In addition to PD-1+ T cell depletion, the modification of p21/p53 signaling in T cells might have the potential to suppress T-cell immunosenescence. Any interventions for the purpose of reducing DNA damage, such as the suppression of oxidative stress in the circulating environment, might become a different approach to reduce T-cell immunosenescence. Therefore, we added the findings of H2AX analysis in the revised Fig. 2D (Fig. 2 legend, P8, L242-L246) and the results (P7, L228-L232), and this assessment in the Discussion (P12, L394-L407).

#2. In Figure 3 and 5, could you also show the total number of CD4 and CD8 T cells in aged mice or under CR condition? The reviewer also would like to know the percentage of Naïve T cells under CR condition.

Thank you very much for your constructive advice. In accordance with the reviewer’s suggestion, we have represented the total number of PD-1+ CD4+ and PD-1+ CD8+ T cells (per g of eVAT) in Figs. 3, 5, and 7. Thus. we show both the total number of PD-1+ CD4+ and PD-1+ CD8+ T cells (per g of eVAT) and the percentage of PD-1+ CD4+, and PD-1+ CD8+ T cells (% of total CD4+ and CD8+ T cells) in Figs. 3, 5 and 7. We considered whether we should also determine the number of CD4+, and CD8+ T cells. The total number of CD4+ and CD8+ T cells is estimated by dividing the total number of PD-1+ CD4+, and PD-1+ CD8 T cells by the ratio of PD-1+ CD4+, and PD-1+ CD8+ T cells, respectively. The change in the number of CD4+ T cells and CD8+ T cells by CR seems to be interesting data, but there may be overlap with the presentation data. In addition, we discussed the changes in CD4+ and CD8+ T cells by CR based on the literature. Therefore, we decided to show both the total number and percentage of PD-1+ CD4+, and PD-1+ CD8+ T cells in Figs. 3, 5, and 7. 

Unfortunately, the number of immune cells obtained from the eVAT of CR mice was relatively low and the number of CR mice was also limited. We divided these immune cells into three tubes for the analysis of macrophages, CD3+ T cells, and regulatory T cells (Treg) (in the present study, we did not show the results of Treg analysis). Therefore, we could not evaluate the expression levels of CD62L in PD-1+ CD44high CD3+ T cells by FACS in CR mice. Therefore, we did not have data regarding the percentage of naïve T cells in the same series of experiments described in Fig. 5. Because CR mice cannot be obtained easily in Japan, we would like to ask the reviewer to understand that we will show the change in the percentage of naïve T cells by CR at the next opportunity.

#3. In Figure 6 and 7, could you show the comparison of each parameter before and after PD-1 depletion antibody? These additional experiments will teach us the potential (prevention or improvement) of senolytic therapy.

According to the referee’s suggestion, we compared the changes in each parameter before and after either control IgG or ant-PD-1 antibody treatment in aged mice. Finally, we did not find any changes in BW and food intake before and after treatment (Figs. 6A and 6B). In addition, anti-PD-1 antibody treatment significantly improved glucose intolerance and insulin resistance, although control IgG treatment did not affect the results of the oGTT and ITT (Figs. 6D and 6E). We also confirmed the effective removal of PD-1+ T cells, especially PD-1+ CD4+ T cells, from the eVAT and spleen (Figs. 7C-F). Therefore, we speculate that the effective depletion of senescence-related T cells was responsible for the increase in naïve T cells in eVAT and the spleen of aged mice (Figs. 7A and 7B). These results strongly suggest that senolytic therapy against senescence-related T cells can lead to the rejuvenation of T cells in aged mice. We have added some sentences in the results, Fig. 6 and 7 legends, and the Discussion in the revised manuscript (P9, L316-L318; P10, L339-L343; P10, L353-P11, L358; P11, L372-L375; P12, L414-L415; and P13, L462-L463, respectively).

Thank you again for your excellent suggestions.

---

## [Decision Letter · Decision Letter 1]

18 May 2021

The effect of caloric restriction on the increase in senescence-associated T cells and metabolic disorders in aged mice

PONE-D-21-02437R1

Dear Dr. Shinmura,

We’re pleased to inform you that your manuscript has been judged scientifically suitable for publication and will be formally accepted for publication once it meets all outstanding technical requirements.

Kind regards,

Ichiro Manabe

Academic Editor

PLOS ONE

Additional Editor Comments (optional):

Reviewers' comments:

Reviewer's Responses to Questions

**Comments to the Author**

1. If the authors have adequately addressed your comments raised in a previous round of review and you feel that this manuscript is now acceptable for publication, you may indicate that here to bypass the “Comments to the Author” section, enter your conflict of interest statement in the “Confidential to Editor” section, and submit your "Accept" recommendation.

Reviewer #1: All comments have been addressed

Reviewer #2: All comments have been addressed

2. Is the manuscript technically sound, and do the data support the conclusions?

Reviewer #1: Yes

Reviewer #2: Yes

3. Has the statistical analysis been performed appropriately and rigorously? 

Reviewer #1: Yes

Reviewer #2: Yes

4. Have the authors made all data underlying the findings in their manuscript fully available?

Reviewer #1: Yes

Reviewer #2: Yes

5. Is the manuscript presented in an intelligible fashion and written in standard English?

Reviewer #1: Yes

Reviewer #2: Yes

6. Review Comments to the Author

Reviewer #1: (No Response)

Reviewer #2: This manuscript is well revised, and all comments have been fully addressed.. I have no further comment.

7. PLOS authors have the option to publish the peer review history of their article (what does this mean?). If published, this will include your full peer review and any attached files.

Reviewer #1: **Yes: **Takuya Chiba

Reviewer #2: **Yes: **Hironori Nakagami

---

## [Editor Report · Acceptance letter]

10 Jun 2021

PONE-D-21-02437R1 

The effect of caloric restriction on the increase in senescence-associated T cells and metabolic disorders in aged mice 

Dear Dr. Shinmura:

I'm pleased to inform you that your manuscript has been deemed suitable for publication in PLOS ONE. Congratulations! Your manuscript is now with our production department. 

Kind regards, 

on behalf of

Dr. Ichiro Manabe 

Academic Editor

PLOS ONE